# Hepatotoxicity in Cancer Immunotherapy: Diagnosis, Management, and Future Perspectives

**DOI:** 10.3390/cancers17010076

**Published:** 2024-12-29

**Authors:** Alberto Savino, Alberto Rossi, Stefano Fagiuoli, Pietro Invernizzi, Alessio Gerussi, Mauro Viganò

**Affiliations:** 1Department of Medicine and Surgery, University of Milano-Bicocca, 20126 Milano, Italymvigano@asst-pg23.it (M.V.); 2Gastroenterology, Hepatology and Transplantation Unit, ASST Papa Giovanni XXIII, 24127 Bergamo, Italy; 3Centre for Autoimmune Liver Diseases, Division of Gastroenterology, Fondazione IRCCS San Gerardo dei Tintori, ERN-RARE LIVER, 20900 Monza, Italy

**Keywords:** immunotherapy, hepatotoxicity, immune checkpoint inhibitors, drug-induced liver injury, immune-related hepatitis, immune-related cholangitis

## Abstract

The growing use of immune checkpoint inhibitors (ICIs) has led to an increased need for managing their associated toxicities, including hepatotoxicity, which is one of the most common immune-related adverse events (irAEs) with potentially serious consequences. The effective management of immune-related hepatotoxicity is crucial to prevent complications and allow patients to continue their oncological therapies. However, current guidelines rely on limited data, often derived from retrospective studies, case reports, and case series, leaving considerable uncertainty regarding optimal diagnostic and treatment strategies. This manuscript offers a comprehensive overview of immune-related hepatotoxicity, from standard care practices to the latest evidence, with the aim of providing clinicians with valuable information to effectively manage these adverse events.

## 1. Introduction

Immunotherapy refers to the manipulation of the immune system aiming to enhance the host’s immune response against tumour cells [1]. It has significantly improved the outcome of oncological treatments over the last few decades and is currently considered the standard of care for several type of malignancies [2,3].

Various cancer immunotherapies have been developed, with immune checkpoint inhibitors (ICIs) being among the most effective and currently widespread. ICIs are monoclonal antibodies that target specific proteins on the surface of immune or tumour cells—namely, cytotoxic T-lymphocyte-associated protein 4 (CTLA-4), programmed death-1 (PD-1), programmed death-ligand 1 (PD-L1), and, recently, lymphocyte-activation gene-3 (LAG-3)—that normally act as brakes on the immune response to prevent excessive inflammatory response or autoimmunity [4,5]. ICIs are approved for several solid and non-solid tumours, including melanoma, lung cancer, hepatocellular carcinoma (HCC), and Hodgkin lymphoma [6].

Despite its effectiveness, ICI therapy is associated with a substantial rate of side effects in many organs, identified as irAEs, with hepatotoxicity being one of the most frequent manifestations [7]. Immune-related hepatotoxicity classically manifests as hepatitis, but, although less frequently, can present also as cholangitis or mixed forms. IrH is defined by prevalent hepatocellular damage with increased serum transaminases and/or bilirubin levels, whereas immune-related cholangitis (irC) is an inflammation predominantly involving the bile ducts and typically characterized by elevations of cholestasis markers.

Given the progressive diffusion of ICI therapies during the last decade and the increasing number of combined schemes already licensed or under study, a tailored management of their toxicity is expected to become highly needed in the near future [8,9].

This narrative review explores the key aspects of hepatotoxicity across ICI treatment. It covers underlying pathological mechanisms, clinical presentations, and diagnostic findings and provides a comprehensive overview of the management landscape, from standard care practice to the latest findings and specific settings.

## 2. Epidemiology

According to a recent, worldwide register, irH accounts for about 8% of all irAEs, with a pooled incidence of all grades around 12.9% [7]. In recent years, the incidence of irH episodes has noticeably increased, likely due to more widespread usage, rising from 8.2% before 2017 to 11% in 2022 [7,10,11]. Although most of the cases are of mild or moderate severity, the incidence of severe irH is not negligible and reported around 6% [10,12]. Death from irH is a potential event with a reported rate of up to 5.2%, even if remains unclear whether all the fatalities registered in the international database were directly attributable to irH progressing to acute liver failure or were related to the underlying malignancy or other comorbid conditions [7]. Notably, the mortality rate has significantly decreased starting from 2018, likely due to improvements in management strategies [7,13,14].

Conversely, IrC is much less common than irH, accounting for only 0.7% of all irAEs, with an overall incidence reported above 1% [7,15,16]. Death from irC is a rare event, with an estimated rate of approximately 4.5% [7].

## 3. Pathophysiology

Immune checkpoints are negative regulators of T-cell activation and play a central role in maintaining the body’s immune balance, preventing the activation of T-cells against self-antigen. Tumour cells, expressing and up-regulating immune checkpoints, exploit this mechanism to escape from the immune system. ICIs counteract this escape mechanism enabling robust T-cell activation and effective immune responses. However, this enhanced immune activity can result in the loss of peripheral tolerance towards some antigen cells, including hepatocytes [17].

The exact pathogenesis of ICI-related hepatotoxicity is not fully understood. Several factors seem to be involved, and the unique immunotolerance of the liver, secondary to the high antigen load of the portal circulation, plays a crucial role. One of these immunotolerant mechanisms lies in the expression of CTLA-4 on regulatory T-cells and PD-L1 on hepatocytes and other non-parenchymal cells, such as stellate and Kupffer cells [18]. ICIs, acting on these pathways, can disrupt this delicate balance and lead to the over activation of T-cells, making the liver vulnerable to acute inflammatory damage [19,20].

CD8+ and CD4+ cell alterations have been widely described in irH, particularly the clonal expansion of cytotoxic, helper (Th)1, and Th17 and the suppression of regulatory (Treg) T-cells [21]. These changes can directly or indirectly cause cell damage through several pathways [21]. Primarily, strong cytotoxic T-cell activity causes injury by releasing effector molecules such as granzyme and perforin [22]. At the same time, increased pro-inflammatory (e.g., interleukin (IL)-12, IL-17, and interferon γ (IFN-γ)) and decreased anti-inflammatory cytokine levels (e.g., IL-10, IL-35, transforming growth factor (TGF-β)) modulate innate immunity, recruiting natural killer cells and macrophages [23]. Also, circulating monocytes are recruited to the liver with increased differentiation into macrophages [24].

In conclusion, ICIs appear to induce changes in the inflammatory environment of the liver through multiple pathways, and both innate and adaptive immunity appear to be engaged in determining cell injury and death. The reason why some patients develop irH while others do not remains unclear. Genetic factors may play a role and, indeed, in a genome-wide association study of 1751 patients on ICIs, an association between three specific loci and the development of irAEs was actually found [25].

The understanding of irC is still not as clear as that of irH. As suggested by Doherty et al., biliary injury may stem from either the recognition of auto antigens on the biliary epithelium or the identification of bacterial epitopes within a colonized biliary tract [26]. Interestingly, the activation of T-cells by biliary epithelial cells is partially dependent on the PD-1 ligand [26]. Further studies are needed to clarify the pathogenic mechanism above immune-related toxicity, particularly irC.

## 4. Risk Factors

Several factors influence the development of irH, including the type and dose of ICIs, underlying malignancies, and individual patient’s conditions and comorbidities.

Prior ICI cycles, combined regimens, longer ICI treatment, and high-dose schemes are associated with a higher hepatotoxicity risk [7,27,28,29]. A recent network meta-analysis reported the least safe regimens as CTLA-4 inhibitors plus chemotherapy (CT) followed by PD-1/PD-L1 inhibitors plus CT [28,30]. Notably, combined regimens with the new LAG-3 inhibitors might be safer, as reported in a preliminary study in which Relatlimab plus Nivolumab showed a global better safety profile compared with Ipilimumab plus Nivolumab [31]. Overall, CTLA-4 inhibitors seem associated with a higher risk of irH compared with PD-L1/PD-1 inhibitors, with Atezolizumab being the safest and Ipilimumab the least safe [11,30,32,33]. The increased degree of injury observed in combined regimens suggests that ICIs themselves may sensitize the liver to drug-induced injuries [34,35].

Regarding underlying malignancies, irH risk is higher in melanoma, renal, and thymus cancers [7,36,37]. The presence of an underlying liver disease plays a significant role: indeed, liver metastases, primary liver cancers (i.e., HCC), and pre-existing chronic liver diseases increase the likelihood of developing irH, according to some reports [38,39,40,41]. However, data are contradictory, since other studies have found no correlation among liver metastases, HCC, and irH rates [42,43,44].

Among patient individual features, pre-existing autoimmune diseases, Asian ethnicity, and female sex seem to predispose to irH [44,45,46]. There is also evidence that a higher risk of irH correlates with a poorer Eastern Cooperative Oncology Group (ECOG) performance status and previous irAEs of any type [7,29,47]. Recent studies found a significant association between higher pre-treatment aspartate aminotransferase (AST), lymphocyte, and eosinophil count levels and a higher incidence of irH [29,43,48,49]. Finally, gut microbiome and genetic factors, such as certain human leukocyte antigens’ alleles (i.e., DRB10301 and DRB107), might play a role in the predisposition to toxicity. However, data are scarce so far, and further evidence is mostly needed [50,51,52]. Similarly, little evidence has suggested a possible correlation between irH and concomitant drugs, such as acetaminophen and proton pump inhibitor use [53,54].

About IrC, to the best of our knowledge, no clear risk factors have been identified at present, due to the scarcity of data. However, irC has been almost exclusively reported with PD-1/PD-L1 inhibitors and rarely with CTLA-4 inhibitor use [16]. Moreover, in a large registry, an association between irC and gastro-oesophageal cancer was described [7].

## 5. Clinical Presentation, Laboratory, and Radiological Features

In line with non-ICI-related drug-induced liver injury (DILI), three distinct patterns of ICI-related damage have been described so far: hepatocellular, cholestatic/cholangitic, and mixed [54]. In turn, cholestatic/cholangitic damage can be subdivided into small ducts, large ducts, and mixed types [16]. Hepatocellular injury (38–52%) is the predominant pattern observed, followed by cholestatic/cholangitic (19–36%) and mixed (24–29%) patterns of liver injury [16,54,55].

These patterns seem to variably correlate with different histology, laboratory, and radiology findings (Table 1).

### 5.1. Clinical Presentation

The time of onset of irH and irC seems to be different [56]. IrH occurs in up to 70% of cases within 3 months from the therapy’s initiation, with a median time typically between 8 and 12 weeks, and an earlier onset with CTLA-4 inhibitors compared to PD-1/PD-L1 inhibitors [57]. Importantly, delayed hepatotoxicity associated with PD-1 inhibitors may emerge up to 60–90 days after the last dose of therapy, which complicates the diagnosis of irH, particularly when additional agents are introduced during this period [58]. Clinicians should maintain a high index of suspicion for irH in patients presenting with liver dysfunction even after the discontinuation of ICIs. Such delayed effects may be attributed to prolonged immune activation or residual effects of the therapy [59].

Elevations as early as 8 days or after 21 months since starting therapy have been rarely reported [7,57,60].

Conversely, only half of the irC cases manifest within 3 months of initiating ICI therapy, with a possible longer onset of the large duct-type injury [16]. Around 91% of cases are present by 12 months [7], with only few cases reported after 2 years [26,61].

The severity of immune-related hepatotoxicity, particularly irH, ranges from mostly mild to less frequent severe cases with acute liver failure [14]. Classically, irH has been classified using the Common Terminology Criteria for Adverse Events (CTCAE) (see Table 2) [62]. The CTCAE, established by the National Cancer Institute and updated until the fifth version, grades hepatotoxicity based on peak abnormalities in liver biochemistry, including ALT, AST, and bilirubin serum values. This system categorizes liver injury from grade 1, indicating mild enzyme elevations, to grade 5, indicating fatal hepatotoxicity. This system is widespread in oncology clinical trials and it is considered the reference standard for assessing immune liver hepatotoxicity, being used by all principal guidelines. Another classification, more familiar to hepatologists, but less used by other, non-liver specialists, is the Drug-Induced Liver Injury Network (DILIN) severity index (Table 2). It considers a wider range of elements, from clinical to laboratory perspectives, including symptoms, hospitalization causes, total bilirubin, international normalized ratio (INR), and others, providing a more global evaluation [63].

From a clinical point of view, a wide heterogeneity of presentations can be observed. Most patients are asymptomatic, while others experience signs and symptoms that are mainly unspecific such as fever, fatigue, general discomfort, abdominal pain (mainly in the upper right quadrant), and loss of appetite [64]. In more severe cases, patients may present jaundice, pruritus, abdominal distension due to ascites, and symptoms and signs indicative of acute liver failure [23].

### 5.2. Laboratory Findings

Routine laboratory evaluations play a crucial role in the differential diagnosis and prognostic evaluation of patients with ICI-related liver toxicity. Baseline liver function tests (LFTs) should be assessed before each ICI infusion, and altered LFTs mandate further investigations.

Elevations in ALT and AST with or without abnormal serum bilirubin are constant in irH and indicate a hepatocellular injury type. Alkaline phosphatase (ALP) and gamma-glutamyl transpeptidase (yGT) values may be elevated in a concomitant cholestasis injury or mixed forms [16,55,65]. In the most severe cases, high levels of bilirubin and a worsening of the international normalized ratio (INR) can occur, too, suggesting the development of acute liver failure [66,67,68].

No specific autoantibodies have been described in irH, in contrast with other irAEs [69]. Similarly, no significant association has been found with other biochemical features typically related to autoimmune-like liver injury such as elevated type G immunoglobulins (IgG) or increased gamma globulins [70].

### 5.3. Radiological Examinations

The role of radiology is mainly ruling out other causes of liver damage, primarily biliary obstructions, cancer progression, or liver steatosis [71,72,73,74]. Radiological exams are also essential to stage liver fibrosis, to rule in or out advanced liver fibrosis/cirrhosis, with a significant impact on clinical management.

In irH, abdominal ultrasound (US) is the initial imaging modality, primarily used to rule out biliary obstruction and to assess liver parenchyma. Steatosis, hepatomegaly, periportal and gallbladder oedema, perihilar lymph nodes, and starry sky pattern (i.e., a heterogeneous liver parenchyma echogenicity with small hyperechoic foci) have been described [75,76,77,78]. Advanced imaging techniques, such as magnetic resonance imaging (MRI) and computed tomography (CT), can provide more detailed evaluations of liver morphology and vascular/biliary structures. Mild to moderate cases usually show normal liver imaging, whereas, in more severe forms, findings include hepatomegaly, periportal and gallbladder oedema, periportal lymphadenopathy, perihepatic ascites, heterogeneous parenchymal enhancement with low-attenuation areas on CT, and periportal T2-hyperintensity on MRI [79,80,81,82]. Notably, some of these findings, such as areas with decreased attenuation, may mimic liver metastases, while others, such as hepatomegaly and periportal lymphadenopathy, can normalize after corticosteroid (CS) administration [75].

Regarding irC, contrast-enhanced MRI with cholangiopancreatography offers the most comprehensive evaluation of the biliary structures [44]. Bile duct injury is mainly defined by non-obstructive dilatations and/or stenosis [54]. Other radiological findings are bile duct wall thickening, enhancement, and irregularity due to inflammation. Bile ducts involvement can be segmental or diffuse, affecting intrahepatic, extrahepatic, or both [16]. Modifications in adjacent structures, such as gallbladder oedema and wall thickening, and Gleason’s sheath edema may be observed too [16].

Few studies have also explored positron emission tomography (PET)/CT imaging in immune-related hepatotoxicity, revealing increased liver fluorodeoxyglucose (FDG) uptake with a higher liver-to-blood-pool standardized uptake value (SUV) mean ratio in irH and increased FDG uptake in the gallbladder and bile ducts, even before the rise in symptoms, in irC [78].

### 5.4. Liver Biopsy and Histological Features

The role of liver biopsy in ICI-induced toxicity is a matter of debate. On one hand, liver biopsy is helpful in challenging cases to better understand the extent and nature of the liver injury, excluding alternative diagnoses [15,47,83,84]. However, on the other hand, histological features are heterogeneous without pathognomonic elements; moreover, in most cases, histology does not impact management choices and might even delay treatment initiation [85]. For these reasons, its use is currently limited to atypical presentations or cases refractory to corticosteroids (CS) [83,84].

IrH predominantly presents with lobular damage of different degrees (typically centrilobular, less often zonal or pan lobular with or without zone 3 accentuation), periportal inflammation (usually mild and less frequently moderate), lobular spotty or confluent necrosis, endotheliitis, and granulomas (loose, well-formed, or fibrin ring type) [56,86,87,88]. Additional findings include mild to moderate steatosis, confined to areas of lobular inflammation and injury, and acidophilic bodies [56,87,88,89]. The inflammatory infiltrate mainly comprises mononuclear cells (i.e., histiocytes and lymphocytes, with or without eosinophils). Less frequently, scattered neutrophils or plasma cells can be present [87]. At immunohistochemistry, CD3+ or CD8+ T-lymphocytes are predominant [86,90].

Some histological findings correlate with the type of ICI. CTLA-4 inhibitors are more commonly associated with confluent necrosis, lympho-histiocytic granuloma-like infiltrates, macrophage aggregates forming microgranulomas, and central vein endotheliitis [56,75,91,92]. In contrast, PD-1/PD-L1 inhibitors often show lobular, non-granulomatous hepatitis with periportal inflammation [87]. Granulomas tend to be more frequent in cases of combination therapy [56,88,93].

Compared to autoimmune hepatitis (AIH), in irH, the periportal inflammation and plasmacytosis are milder and there is a higher proportion of CD8+ cells [55,82,86,90,94].

Histologically, irC is characterized by absent or only focal lobular injury, mild to marked bile duct injury and loss, and portal inflammation with mononuclear or mixed infiltrate [16,75,86,87,95,96,97]. These features resemble conditions like DILI and primary sclerosing cholangitis [65]. Cases of biliary duct absence mimicking vanishing bile duct syndrome have been rarely reported [98,99,100]. Extrahepatic bile duct involvement can occur with pathology samples characterized by constant inflammatory infiltration in the lining epithelium and, in a third of cases, by diffuse fibrosis [16].

## 6. Differential Diagnosis

As discussed above, there are no pathognomonic findings of ICI-related hepatotoxicity, so the diagnostic work-up involves a comprehensive evaluation of medical history and clinical, laboratory, and imaging features to accurately characterize the condition.

Above all, it is important to remember that, before starting ICI therapy, a routine screening is mandatory to assess the risk of developing irH and discovering an unknown underlying liver disease [15,83,84,101]. Current guidelines recommend testing for total serum bilirubin, AST, ALT, yGT, and ALP levels at baseline [82]. Screening for hepatitis B virus (HBV), hepatitis C virus (HCV), and human immunodeficiency virus (HIV) infections is also required at baseline [83].

After the initiation of ICI treatment, once elevated LFTs are identified, the first, crucial step is to exclude alternative aetiologies of liver damage. Indeed, in clinical trials, only 20–30% of all patients with LFTs have received a definitive diagnosis of irH [102,103], and, in a recent study, more than 60% of patients presenting a cholangitis pattern of injury had indeed competing causes [85].

As patients with cancer have a wide range of potential alternative aetiologies, a comprehensive assessment is essential to exclude viral infections, autoimmune diseases, infiltrative tumour spread, biliary diseases, vascular abnormalities, iron overload, and DILI, including exposure to dietary supplements or herbal products [15]. An important point concerns distinguishing irH from AIH. IrH typically lacks specific autoantibodies, does not show IgG elevation, and shows different histological features, as reported previously. Riveiro-Barciela et al. also found slight differences in clinical presentation, with AIH patients being younger and complaining of more symptoms (such as fatigue and pruritus) than those with irH [70]. Similarly, higher autoantibodies’ presence (i.e., anti-nuclear antibodies (ANA), perinuclear anti-neutrophil cytoplasmic antibodies (p-ANCA), and anti-smooth muscle antibodies (ASMA)) were more represented in AIH compared with irH [68]. Also, CS responsiveness is usually more evident in AIH, with irH often requiring higher CS doses [104].

Similarly, irC needs to be differentiated carefully from other causes of intrahepatic and extrahepatic cholestasis, such as other immune-mediated cholangiopathies (i.e., primary biliary cholangitis, primary sclerosing cholangitis, and IgG4-related cholangitis), malignant disorders (especially gastrointestinal cancers), infections, and other drug-induced biliary lesions [16,105,106,107,108]. This distinction may be difficult in clinical practice; for instance, periductal-infiltrating cholangiocarcinoma and metastasis may present a growth along the bile ducts, mimicking that of irC [16].

Finally, to assess the likelihood that a certain drug causes a liver injury, it can be helpful to calculate the Roussel Uclaf Causality Assessment Method (RUCAM) [109]. This scale includes seven domains (i.e., the timing of the liver injury and outcome of de-challenge, other potential causes, concurrent drug use, risk factors, historical data, and the re-administration response), and scores range from −10 to +14, categorizing the probability of DILI. Recently, the Revised Electronic Causality Assessment Method (RECAM) scores, a semi-automated, computerized, and validated scoring system modified from RUCAM was developed, showing similar performance [110].

## 7. Treatment Strategies

All current guidelines recommend a step-up approach for managing immune-related hepatotoxicity, based on the severity grade as classified by the CTCAE 5.0 criteria (Figure 1) [64,83,84,111,112].

It is always necessary to withhold any potential liver toxic medication and alcohol intake and to closely monitor biochemically and clinically, with increasing frequency ranging from 1–2 times per week (G1) to daily assessment for more severe forms (G ≥ 3) of irH, respectively. Hepatology consultation is recommended for >G2 liver injury and for more challenging cases. Inpatient monitoring is safer when transaminase levels are ≥8 × the upper limit of normal (ULN) and/or total bilirubin levels are >3 × ULN. Supportive care is needed in symptomatic patients, and transfer to a tertiary care facility might be considered individually.

Overall, treatment strategies lie in ICI temporary or permanent discontinuation and CS use with increasing doses depending on the severity of liver injury. For G1 irH, ICI therapy can be continued, while in G2 cases, the treatment should be temporarily withheld. If there is no improvement in LFTs after 3–5 days, CS therapy (0.5–1 mg/kg/day) should be considered. In this case, CS tapering may start when serum levels improve to G1 and should be maintained for at least 1 month. ICI treatment can be resumed when the CS dose is ≤10 mg/day and LFTs have recovered to G1 [86]. For more severe irH (≥G3), the permanent discontinuation of ICIs is not required in all cases, but it is mandatory for symptomatic patients and G4 episodes. In this setting, the prompt initiation of CS treatment at higher dosages (1–2 mg/kg/day) represents the standard of care. CS tapering can be attempted at 4–6 weeks when LFTs have improved to G1, although the optimal timing is still unclear [85,87]. CS treatment results in an effective response in up to 80% of cases, but steroid resistance or refractoriness, defined by partial or absent LFT improvement within 3 days from CS initiation, is a significant event with reported incidence ranges between 12% and 36% [41,113,114,115,116,117,118]. In this context, additional immunosuppressive therapies should be considered [113,118]. However, neither optimal posology nor specific second line schedules have been established due to the lack of evidence, supported mainly from case reports and series [64,83,84]. Disease severity, ICI type, kinetics of LFTs’ rise, and concurrent metastatic disease are other factors to be considered for adding a second-line immunosuppressor [119]. In a recent systematic review by Hwang et al., 75% of patients with irH received steroid treatment, while 16% of them required a secondary immunosuppressive drug [120].

After the introduction of a second-line immunosuppressive drug, up to 90% of the patients show a positive outcome [70,121]. Mycophenolate mofetil (MMF), an antimetabolite immunosuppressant, is a relatively safe and effective choice, with more evidence available on its use in this setting. It is important to consider that the clinical effect of MMF is not immediate, as it requires time to modulate the immune response. Indeed, the achievement of steady-state plasma concentrations of its active metabolite, mycophenolic acid (MPA), typically occurs within 48–96 h with regular dosing [122,123,124,125]. Also, calcineurin inhibitors, such as tacrolimus and cyclosporine, and other antimetabolites such as azathioprine have shown some positive results [116,117,121,126,127,128,129,130,131]. Anecdotal reports suggested tocilizumab, an IL-6 receptor-neutralizing antibody, and basiliximab, an anti-CD25 monoclonal antibody, as promising options [119,132,133]. Similarly, the use of anti-thymocyte globulin and plasma exchange in severe, fulminant irH have been reported with positive outcomes [134,135,136,137]. Notably, the use of infliximab, a tumour necrosis factor-alpha blocker, is not advised due to concerns about liver toxicity; however, several reports have shown beneficial and safe effects [121,138,139,140,141,142].

Concerning irC, less evidence is available. Although CSs seem to be less effective for the treatment of irC with a high percentage of null- or partial-responders, they may be beneficial in reducing inflammatory infiltration in the early stage [16,26,143]. A second-line immunosuppressive drug might be considered in the most severe cases [144]. Ursodeoxycholic acid (UDCA), a safe, well-tolerated drug with cytoprotective, antiapoptotic, and immunomodulatory effects on bile duct epithelium, is suggested [83,145]. The treatment response typically shows a slow decrease in cholestasis indices, with abnormal biliary enzymes still at 18 months, suggesting long reparative processes [16,146].

Few data exist about the use of other medications. Bezafibrate or n-acetylcysteine have been hypothesized to help or hasten the normalization of LFTs [132,147]. Also, budesonide as a steroid-sparing strategy for longer therapy durations was assessed with positive results [148,149]. Given the little data available about these additional drugs, large-scale validation is needed and there are no strong recommendations for their use.

Regev et al. [150] provided a comprehensive review of irH caused by ICIs under development during clinical trials, giving some recommendations for management in this specific setting. For instance, they recommended that patients with serum ALT > 3 × ULN be excluded from ICI clinical trials unless HCC metastatic disease is present. In the presence of hepatic primary or metastatic disease, it is recommended to exclude patients with serum ALT > 5 × ULN.

## 8. Impact of Immune-Related Hepatotoxicity and Immunosuppressive Therapy on Cancer Response and Patient Survival

All current guidelines recommend a step-up approach for managing immune-related hepatotoxicity, based on the severity grade as classified by the CTCAE 5.0 criteria (Figure 1) [64,83,84,111,112].

### 8.1. Immune-Related Hepatotoxicity and Patient Survival

The occurrence and severity of irAEs, including immune-related hepatotoxicity, seem to be associated with better tumour responses, potentially reflecting enhanced and more effective immune activation against tumours [46,121,151,152,153]. Several studies have described that patients who experienced irAEs have significantly longer overall survival compared to those who do not, with irAEs identified as an independent factor of better prognosis [43,47,154,155,156,157]. Indeed, discordant results have been reported in this setting, highlighting the importance of considering potential confounding factors, above all, the possibility that patients with longer survival may be affected by a higher incidence of irAEs due to prolonged exposure to ICIs. Interestingly, some authors have reported less or no survival benefit in those who had developed immune-related hepatotoxicity, especially early and severe forms, compared to other irAEs [158,159,160,161]. More data are needed to confirm irAEs and, specifically, hepatotoxicity as surrogate indicators of ICI efficacy [162,163].

### 8.2. Re-Challenge ICI Therapy

A critical clinical dilemma regards the decision to permanently discontinue ICIs after the development of severe irH (≥G3), as recommended by guidelines [83,84]. Growing evidence suggests that retreatment with ICIs after hepatotoxicity may be a feasible and relatively safe option in selected cases [164,165].

Hountondji et al. found a recurrence of irH in 23.5% of patients retreated with ICIs who had previously developed ≥G3 irH [54]. Similarly, Riveiro-Barciela et al. reported in a prospective study that 65% of patients with severe irH (≥G3) well tolerated ICI retreatment and only 34.8% of them, at the time of retreatment, received a concomitant low-dose (5–10 mg) CS [152]. In a retrospective study by Li et al., 48% of patients with previous irH experienced recurrent irH, but only 19% required discontinuation, indicating a relatively modest risk of severe relapse [166]. In another study, none of the 12 patients with previous ≥ G3 irH experienced irH recurrence after a median follow-up of 9.3 months, with only a mild elevation of liver injury tests in three cases [137]. In a systematic review, among patients with previous irH (40%) who restarted ICI, irH recurrence was documented in 22% of cases, mostly with less severe presentations than the index event [120].

Risk factors for irAEs’ recurrence after ICI retreatment include an immunological background (i.e., autoimmune disease history or autoantibodies’ positivity), anti-CTLA-4 regimen, and younger age. Moreover, patients with hepatocellular patterns of injury appeared to have a better tolerance of ICI reintroduction or re-challenge compared to those with mixed or cholestatic patterns [137]. Overall, these findings support the feasibility of ICI retreatments after severe irH or other irAEs, at least in selected patients.

### 8.3. Side Effects and Impact of CS and Other Immunosuppressors on Patients with Cancer

The impact of immunosuppressive therapy for irAEs on cancer response and patient survival is a matter of concern. Theoretically, since ICIs act on T-cells through checkpoint inhibition, broad immunosuppression may reduce their effects. Overall, available data suggest that the use of CS for irAEs may not have a large deleterious effect on overall survival [167]. However, there is no specific data on hepatotoxicity alone, and contrasting results have been reported.

A systematic review reported that CS use in patients with irAEs did not influence the overall survival [70] as well as the timing of CS administration (within 30 days vs. after 30 days) from irAEs [158,168]. However, other studies showed a possible detrimental effect on survival of early CS administration and high CS peak dose [158,169,170,171].

Regarding other immunosuppressors, clear evidence is lacking to date. Specifically, few studies have reported contrasting results about tumour necrosis factor-alpha blockers and the use of concomitant second-line agents [172,173,174]. MMF showed good safety in the study by Alouani et al., where there was no difference in overall survival and progression-free survival in patients treated with MMF plus CS and CS alone [175]. Similarly, tocilizumab use appeared safe in a review including 91 patients with no reports of disease progression [176].

Finally, regarding older patients, ICIs could be safely administered also in them with comparable survival outcomes to younger individuals, although evidence is still scarce [177].

## 9. Specific Setting

The presence of an underlying liver disease influences the differential diagnosis and the therapeutic approach to irH. Herein, we discuss some specific issues arising when ICIs are used to treat hepatobiliary cancers and in chronic hepatitis B where the interplay between ICI therapy and HBV increases the risk of HBV reactivation (HBVr).

### 9.1. Hepatocellular Carcinoma

HCC is a histological subtype of primary liver cancer, which accounts for more than 80% of all hepatic cancers. It is the sixth most commonly diagnosed cancer and the third most common cause of cancer-related death globally [178,179]. Systemic therapy for unresectable HCC (Barcelona Clinic Liver Cancer (BCLC) stage C) lies in combination regimens of ICIs (Durvalumab + Tremelimumab) or ICI + anti-vascular endothelial growth factor (Atezolizumab + Bevacizumab), currently superior to tyrosine kinase inhibitors [180,181,182,183]. The number of targeted therapies and combination schemes is likely to increase further, given the high number of ongoing trials [184]. Moreover, an increasing number of studies are exploring the potential role and timing of ICIs for HCC, as neo-adjuvant or adjuvant either for surgical or loco-regional treatments [185,186].

The incidence of all grades and types of irAEs was reported as similar to other cancer settings, rating approximatively 34% [187,188]. Moreover, irAEs were mostly well tolerated, with only 18% of patients requiring CS therapy and 6% therapy discontinuation [187]. Similar results were reported in a study focused on low-socioeconomic areas of the United States, thus adding to the safety profile of ICI treatment [189]. Hepatotoxicity is one of the most common irAEs in HCC treatments. Data reported that there is a higher risk of hepatotoxicity, especially ≥G3, in patients treated with ICIs for HCC (11.5%) compared with other tumours (2.6%), with a 10-fold higher incidence and earlier onset of hepatic injury [38,186,190,191,192]. However, resolution of irH occurred in up to 80% of patients without CS, with a low rate of hepatic decompensation [38,186]. These findings suggest that LFT elevation in this context might not solely indicate hepatic toxicity but could reflect tumour cell lysis or higher T-cell infiltration and inflammation, raising concern about the role of aminotransaminase level as the sole criteria for assessment in this setting.

As discussed above, the hypothesis that patients experiencing irAEs might show better survival rates is still controversial. Indeed, Celsa et al. reported longer overall survival in patients with G1/2 irH compared to those who did not develop irH among HCC ICI treatments [38]. Conversely, Tian et al. found no benefit on overall survival but only a prolonged progression-free survival [187].

CTLA-4 inhibitors seem to induce more frequent irAEs, while PD-1 inhibitors seem to be associated with a higher risk of toxicity than PD-L-1 inhibitors [191,193]: this is likely due to the different molecular mechanisms of action, although the exact reason is still unknown. As an example, CTLA-4 inhibitors exert their action in the early stages of immune activation, promoting a more generalized activation of T-cells compared with other ICIs [194], whether PD-1 inhibitors act on both PD-L-1 and PD-L-2 receptors and PD-L-1 affects only the PD-L-1 pathway [194,195].

At present, even if data on re-challenge are still scarce, it appears safe also in the HCC setting. Scheiner et al. reported that around half of patients presented any-grade irH, but only 17% presented a G3 or G4 irH and only 12% required CS therapy at the second ICI cycle. Moreover, only 3% of patients had to discontinue treatment due to toxicity, and no treatment-related deaths were recorded [196]. Similarly, Joerg et al. reported a good safety profile in patients treated with Atezolizumab + Bevacizumab previously exposed to other systemic treatments [197].

### 9.2. Cholangiocarcinoma

Cholangiocarcinoma (CCA) is a malignancy originating from the bile ducts, often presenting a significant therapeutic challenge due to its typically late diagnosis and resistance to conventional treatments. ICIs targeting PD-1, PD-L-1, and CTLA-4 have emerged as a promising treatment option for CCA, with good results in terms of tolerability and safety.

In the TOPAZ-1 trial, a phase III study evaluating the benefit of adding Durvalumab to first-line CT in advanced biliary tract cancer, the incidence of ≥G3 hepatotoxicity was 7.6% in the Durvalumab + CT group vs. 5.3% in the CT group, while the incidence of any ≥G3 irAE was 62.7% and 64.9%, respectively, suggesting that the addition of Durvalumab did not significantly increase the overall incidence of severe AEs compared to CT alone [198]. Another phase III trial (KEYNOTE-966), which assessed the efficacy and safety of Pembrolizumab combined with Gemcitabine and Cisplatin in advanced biliary tract cancers, reported an incidence of transaminase elevations leading to treatment interruption in only 2.5%, with no significant difference in ≥G3 toxicity with standard CT [199]. Finally, in the KEYNOTE-158 trial, which evaluated Pembrolizumab monotherapy across various types of advanced solid tumours including biliary tract cancers, the incidence of irH of any grade was 0.9%, with only 0.3% ≥G3 [200].

Despite the limited data, these results are encouraging, showing a low incidence of severe irAEs including hepatotoxicity. Overall, the risk–benefit profile of ICIs in CCA seems favourable, particularly in specific molecular subgroups, such as those with high microsatellite instability or deficient mismatch repair [200].

### 9.3. Hepatitis B Reactivation

Although ICI could reduce the HBV viral load by blocking the overexpressed PD-1 and CTLA-4 on HBV-specific T-cells, HBVr is a possible and concrete event during ICI therapy [201,202,203,204,205]. In HBsAg-positive patients, the reactivation rate is reported around 2% and 11% in those, respectively, receiving and not receiving nucleos(t)ide analogue (NA) prophylaxis. Instead, in HBsAg-negative/HBcAb-positive patients, HBVr developed in only 0.2%, whether under NA prophylaxis or not. In both cases, liver decompensation and death were not observed.

In conclusion, all patients planned to receive ICI should be screened for HBV before starting ICI therapy; all HBsAg-positive patients should receive antiviral therapy with NUCs, whereas patients with resolved HBV infection should not receive antiviral prophylaxis but should undergo a 3-monthly monitoring of liver transaminase, HBsAg, and HBV DNA [206,207,208].

## 10. Discussion

The use of ICIs is rapidly spreading as they currently represent the standard of care in most malignancies; consequently, the number of irAEs to be managed has progressively increased. Immune-related hepatotoxicity is one of the most common immune-mediated side effects and the knowledge of its framework is essential for a proper treatment management. However, this presents several challenges, particularly concerning the need for an accurate assessment of both pre-treatment liver condition and the ongoing liver injury together with the optimization of treatment strategies. Furthermore, current guidelines provide guidance based on limited data, mainly retrospective or case reports and series, but most recent studies have raised concerns about the possibilities of new approaches, like tailoring treatment based on patient-specific factors, such as liver comorbidities, cancer type, and the severity of irH, or like the development of reliable biomarkers to predict irH risk, guide treatment decisions, and monitor response [15,64,83,84].

Immune-related hepatotoxicity is classically graded by the CTCAE classification. If, on the one hand, this classification provides an easy-to-use and standardized score to assess liver injury, on the other, a lack of concordance between the CTCAE and the clinical severity of irH is not infrequent, with the potential risk of overestimating the severity of the condition and consequent overtreatment. Indeed, CTCAE only considers some elements (ALT, AST, ALP, and bilirubin elevations) without adequately taking into account the clinical aspect of liver function. In contrast, the DILIN score, widely used to grade other DILIs, offers a more comprehensive evaluation by incorporating clinical and laboratory factors, including the parameters commonly used by hepatologists to assess the presence of liver failure (prothrombin time and hepatic encephalopathy). This discrepancy was highlighted in some studies where irH was graded with both CTCAE and DILIN scores. In the study by Hountondji et al., although the majority of irH was CTCAE ≥ G3, the DILIN scored no higher than moderate and none of the patients evoked criteria of severe hepatitis [54]. Similarly, Riveiro-Barciela found a closer relationship between DILIN and irH grade rather than CTCAE, with one case classified as severe in DILIN score that developed acute liver failure, showing a good reliability of this latter score [152]. Overall, these data suggest that adopting DILIN criteria could improve the accuracy of the liver assessments and prompting more appropriate management decisions, first of all, questioning the need to permanently discontinue ICI in every CTCAE ≥ G3 hepatotoxicity event.

Another matter of debate regards the current role of liver biopsy in diagnosis and management of ICI-induced hepatotoxicity. As indicated in the guidelines, a liver biopsy should not be routinely performed but dedicated to the most challenging cases, primarily CS refractory settings. Although liver biopsy is a safe procedure with a low rate of adverse events, most of them of mild entity, it remains an invasive procedure and its associated risk must be counterbalanced by clear advantages in terms of diagnostic yield and management impact [209,210]. Concerns about its utility are enhanced by data reporting a poor correlation with clinical outcomes and treatment response [55,87], other than the possibility of delayed treatment due to the procedure [85]. Conversely, liver biopsy presents some undisputed strengths in differential diagnosis (particularly in irC cases) and in characterizing patterns of liver injury, its severity (which is poorly predictable by the current biochemical profile), and inflammatory infiltrates [55,87]. In this regard, Pi et al. suggested an underestimation of small duct irC due to the scarce use of liver biopsy and Reddy et al. highlighted the benefits of liver biopsy, describing an interesting diagnostic change from irH to irC with relevant therapeutical implications [16,132]. Similarly, the case reported by Kanaoka et al. about delayed ICI hepatotoxicity developed after 2 months from the end of ICI therapy showed the utility of histological evaluation in uncommon presentations [128]. As De Martin et al. suggested, histology can be reevaluated also in more classical cases from the perspective of tailored medicine [56]. Indeed, a better characterization with a methodical approach might guide a more precise treatment, avoiding unnecessary systemic CS and using other immunosuppressors or drugs as first-line choices, such as UDCA in mixed and cholestatic injury patterns [54,55,87]. For example, patients with predominant T-cell infiltrates and interferon-γ or T-helper 1 pro-inflammatory cytokine profiles seem to respond better to T-cell-targeted therapy with the IL-6 receptor-neutralizing antibody [211], whether those with hepatotoxic patterns and granulomatous profiles resulted in better response to CS [16,92]. Moreover, specific histological markers, such as 2,3-dioxygenase 1, which correlates with irC pattern and response to UDCA treatment, might be shortly available in clinical practice [212]. Zeng et al. presented promising data on transcriptomics analyses and correlation among IL-1B overexpression, irH susceptibility, and poorer outcomes [213]. Although there is a clear need for reliable non-invasive biomarkers for liver injury pattern characterization, treatment response, and prognosis, liver biopsy may still cover a wider role in their absence.

Always focusing on precision medicine, the systematic use of the R ratio may be an effortless initial step to guide liver injury characterization and treatment personalization [16,63,214]. Indeed, the rate of cholangitic/cholestatic and mixed pattern injury seems higher than expected with a substantial CS resistance [55,65]. In this sense, UDCA may be more impactful, although guidelines are not fully aligned [54]. Meunier et al. described cases of irC efficaciously treated only with UDCA, showing the marginal role of CS in this specific setting [214]. In conclusion, we propose considering liver biopsy before using second-line therapy, without delaying its start.

Concerning CS, they are the cornerstone of treatment for moderate to severe irH, yet optimal indications, dosing, and duration remain not fully elucidated. Uncertainty about their role has risen primarily from several studies describing significant recovery rates without CS administration (up to 50% in some case series and also in severe hepatitis), faster improvements of untreated than treated irH, and scarce efficacy in cholestatic/cholangitic injury pattern compared to hepatocellular or mixed ones [54,56,137,215]. Some evidence suggests that lower doses may be as effective as higher doses with reduced risk of CS-related adverse events such as infections or hyperglycaemia [121,216]. In contrast, some authors have reported that insufficient initial CS doses with subsequent increases may facilitate the development of CS refractory condition, and higher CS dosage than what is recommended by guidelines may be needed in ≥G4 cases [137,217]. These contrasting data highlight the need to reassess the role of CS in this setting. Probably their use could be reduced and addressed to more selective cases, also in light of the availability other immunosuppressive options. Indeed, in cases with an initial insufficient response to CS, CS de-escalation with the imbrication of a second-line immunosuppressor rather than a CS increase may be more beneficial [121]. Similarly, a transaminase normalization was observed as slower in CS monotherapy than in combination regimens (i.e., CS plus a second immunosuppressor) with a comparable incidence of adverse events [121]. Also, budesonide, a liver-directed topical steroid, may play a role in long-term treatment in lowering systemic CS adverse effects [148,149]. Overall, the role of immunosuppressants appears critical, and the sequence of drug administration used in combination therapy can influence both efficacy and safety [218]. This suggests that studies focusing on the optimization of current schemes are warranted.

The decision to definitively discontinue or retreat patients after the development of severe cases of hepatotoxicity is complex and must balance the risks of recurrent episodes against the potential oncological benefits, even more so in those cases where no other options are available. Although guidelines are quite conservative and suggest permanent withdrawal in severe forms of hepatotoxicity, emerging evidence indicates that some patients may tolerate the reintroduction of ICIs without short- and long-term negative outcomes. Generally, the re-challenge can be pursued with an ICI class switch or with the reintroduction of the same class agent or molecule with or without a concomitant immunosuppressive therapy. Several studies highlighted these re-challenge strategies, showing a good tolerance (up to 70% without hepatotoxicity relapse) and, importantly, a low rate of severe liver damage recurrence even in higher-risk settings, such as HCC patients [60,112,151,163,164,195,196]. Interestingly, Lallement et al. reported a case where ICI therapy was not withdrawn after the rise in irH, questioning the need to suspend the treatments at least in mild/moderate cases [219]. Unfortunately, standardized protocols for retreatment strategies are lacking, and updated guidelines are expected to fill this gap.

In conclusion, optimizing the management of ICI-induced hepatotoxicity requires a multifaceted approach that includes refining assessment tools for liver injury severity, effective use of liver biopsies, tailored immunosuppressive therapies, and consideration of re-challenging ICIs when appropriate. Ongoing research into predictive biomarkers and novel therapeutic strategies will be critical in improving outcomes for patients with immune-related hepatotoxicity, in specific settings such as CCA and HCC.

## 11. Future Direction

The management of irH presents growing challenges. Prospective studies, including large observational registries, randomized controlled trials, and real-world data are essential to improve our understanding of the pathophysiology, identify high-risk patients, and establish evidence-based management strategies. Future research must prioritize the development of robust protocols for irH treatment and stronger recommendations about retreatment after irH. The role of CS, with a particular focus on dose and duration, needs to be further validated as well as their impact on patient survival and cancer response. Identifying non-invasive predictive markers for susceptibility and treatment response is essential in light of personalized medicine. Similarly, the role of Tregs should be better explored in future research as available data have underlined their role in both HCC development and ICIs’ toxicity [220]. Future studies should also include specific forms of geriatric assessment aiming to achieve optimal patient stratification. Furthermore, as ICI use for HCC and CCA is progressively increasing, a better understanding of their safety and efficacy as well as of combination strategies such as loco-regional treatments should be prioritized.

## 12. Conclusions

Immune-related hepatotoxicity, primarily manifesting as irH and, less frequently, as irC, is a significant challenge in the management of patients with cancer receiving ICI therapy, even more so with the expanding use of combination regimens. Most cases are of a mild entity, but the incidence of severe cases is as significant as the possibility of fatal cases. As most patients are asymptomatic or with a nonspecific presentation, the assessment of laboratory, radiological, and histological findings is crucial. While the pathophysiology of ICI-induced hepatotoxicity is increasingly understood but not fully elucidated, substantial gaps remain in predicting which patients are at greatest risk. CTLA-4 inhibitors, combination regimen, previous irAE, and a history of autoimmune disease represent the most recognized risk factors. The proper management lies in close monitoring, early diagnosis, and immunosuppressive therapy in higher-grade or challenging cases, ranging from CS administration to other immunosuppressive therapies. The impact of irAEs on cancer and patient outcomes seems favourable, with better survival in those experiencing irAEs, although hepatotoxicity appears less associated with this benefit survival, particularly in the early and most severe cases. Immunosuppressor use for irAEs does not seem to be associated with worse survival or significant adverse events, and ICI re-challenge after discontinuation may be a feasible option at least in selected patients. Similarly, ICI therapy appears not only effective but also safe in specific settings including HCC, CCA, and HBV-infected patients.

## Figures and Tables

**Figure 1 cancers-17-00076-f001:**
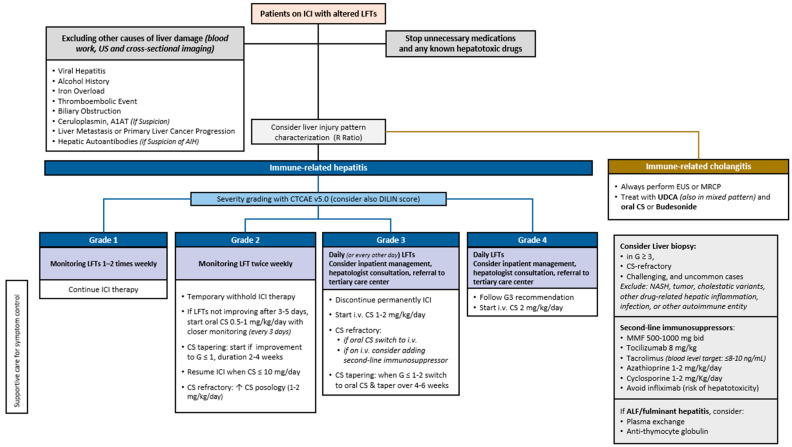
Proposed approach to immune-related hepatotoxicity. Abbreviations: immune-related hepatitis (irH), immune checkpoint inhibitor (ICI), liver function test (LFT), ultrasound (US), alfa-1-antitripsine (A1AT), autoimmune hepatitis (AIH), magnetic resonance cholangiopancreatography (MRCP), endoscopic ultrasound (EUS), ursodeoxycholic acid (UDCA), corticosteroid (CS), Common Terminology Criteria for Adverse Events (CTCAE), Drug-Induced Liver Injury Network (DILIN), mycophenolate mofetil (MMF), increase (↑).

**Table 1 cancers-17-00076-t001:** Main features of liver injury patterns.

		Pattern of Liver Injury	
		Hepatocellular	Cholestatic/Cholangitic
Time of onset from ICI therapy		70% within 3 moths ↑↑↑* AST/ALT ± bilirubin ↑** ALP and/or yGT (mainly in mixed forms)	90% within 12 months (only 50% within 3 months)
Laboratory		↑↑↑ AST/ALT ± bilirubin ↑ ALP and/or yGT (mainly in mixed forms)	↑↑↑ ALP and/or yGT ↑ AST/ALT
Imaging	US	Normal in mild/moderate forms Steatosis, hepatomegaly Periportal and gallbladder oedema Starry sky pattern Periportal lymphadenopathy Perihepatic ascites	Bile ducts dilatation
	CT	Normal in mild/moderate forms The same of US (except starry sky pattern) + heterogeneous parenchymal enhancement with low-attenuation areas	Bile duct dilatation, wall thickening, enhancement
	MRI	Normal in mild/moderate forms The same of US (except starry sky pattern) + perihepatic ascites periportal T2-hyperintensity	Bile ducts irregularity, stenosis, dilatation, wall thickening, enhancement Gallbladder oedema and wall thickening, Gleason’s sheath oedema
	PET/CT	↑ liver FDG uptake	↑ gallbladder and bile ducts FDG uptake
Histology		Lobular damage (mainly centrilobular) Mild/moderate periportal inflammation Lobular spotty or confluent necrosis Endotheliitis Granulomas Mild/moderate steatosis (confined to inflamed areas) Acidophilic bodies	Absent or focal lobular injury Mild/marked bile duct injury and loss (irregularity and degeneration of bile duct epithelium, cytoplasmic vacuolization, periductal fibrosis, cholestasis, intraepithelial lymphocytes infiltration, ductular proliferation, biliary-type interface activity, and neutrophilic pericholangiolitis) Portal inflammation

Abbreviations: Ultrasound (US), computed tomography (CT), magnetic resonance imaging (MRI), positron emission tomography (PET), fluorodeoxyglucose (FDG), alanine aminotransferase (ALT), aspartate aminotransferase (AST), upper limit of normal (ULN), alkaline phosphatase (ALP), gamma-glutamyl transferase (yGT). ↑↑↑* Marked increase, ↑** Mild increase. ↑ increase.

**Table 2 cancers-17-00076-t002:** Grading of hepatotoxicity by the Common Terminology Criteria of Adverse Events (CTCAE) v5 and Drug-Induced Liver Injury Network (DILIN).

Grade	CTCAE	DILIN
**Grade 1**	ALT/AST >1–3 × ULN >1.5–3 × baseline if baseline was abnormal ALP/yGT >1–2.5 × ULN >2–2.5 × baseline if baseline was abnormal Total bilirubin >1–1.5 × ULN >1–1.5 × baseline if baseline was abnormal	ALT and/or ALP elevation above ULN but Total bilirubin <2.5 mg/dL and INR <1.5
**Grade 2**	ALT/AST >3–5 × ULN >3–5 × baseline if baseline was abnormal ALP/yGT >2.5–5 × ULN >2.5–5 × baseline if baseline was abnormal Total bilirubin >1.5–3 × ULN 1.5–3 × baseline if baseline was abnormal	ALT and/or ALP elevation above ULN but Total bilirubin ≥2.5 mg/dL or INR ≥1.5
**Grade 3**	ALT/AST >5–20 × ULN >5–20 × baseline if baseline was abnormal ALP/yGT >5–20 × ULN >5–20 × baseline if baseline was abnormal Total bilirubin >3–10 × ULN >3–10 × baseline if baseline was abnormal	↑ALT, ALP, total bilirubin, and/or INR and hospitalization due to DILI or prolonging existing hospitalization due to DILI
**Grade 4**	ALT/AST >20 × ULN >20 × baseline if baseline was abnormal ALP/yGT >20 × ULN >20 × baseline if baseline was abnormal Total bilirubin >10 × ULN >10 × baseline if baseline was abnormal	↑ ALT and/or ALP + Total Bilirubin ≥ 2.5mg/dL and at least one of the following: ○ Hepatic failure (INR ≥1.5, ascites, or hepatic encephalopathy); ○ other organ failure believed to be due to DILI event
**Grade 5**	Death due to hepatotoxicity	Death or liver transplant due to DILI event

Abbreviations: Alanine aminotransferase (ALT), aspartate aminotransferase (AST), upper limit of normal (ULN), alkaline phosphatase (ALP), gamma-glutamyl transferase (yGT), international normalized ratio (INR), drug-induced liver injury (DILI), increase (↑).

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
