# Peer review of "Hepatotoxicity in Cancer Immunotherapy: Diagnosis, Management, and Future Perspectives"

_cancers, 2024, doi:10.3390/cancers17010076_

Round 1
Reviewer 1 Report
Comments and Suggestions for Authors
This is an interesting review manuscript addressing the key aspects of hepatotoxicity across immune checkpoint inhibitor (ICI) treatment. It is well written and organized. I would suggest a minor point to further improve the manuscript. As ICIs are currently approved for the first-line systemic treatment of hepatocellular carcinoma (HCC) treatment and most HCC patients have an underlying cirrhosis, the development of hepatoxicity is of paramount clinical relevance. Recent studies have reported a crucial role of CD4+CD25+Foxp3+ regulatory T cells (Tregs) in the pathogenesis of ICI-related hepatotoxity as this immune cell population largely expresses molecular targets of ICIs, as previously described (CTLA-4 control over Foxp3+ regulatory T cell function. Science. 2008;322:271–275; doi: 10.3748/wjg.v27.i22.2994; doi: 10.1126/science.1160062; doi: 10.1007/s00262-019-02427-4; doi: 10.1016/j.semcancer.2019.01.006.). Of relevance, Treg frequency and function is a determinant factor affecting the disease-progression of the underlying chronic liver disease as well as the risk of HCC development/spread and ICI-related hepatotoxicity (doi: 10.3748/wjg.v27.i22.2994) and future clinical studies should explore the pattern of regulatory T lymphocytes during treatment to correlate the development of adverse events with Treg temporal trend.
Author Response
This is an interesting review manuscript addressing the key aspects of hepatotoxicity across immune checkpoint inhibitor (ICI) treatment. It is well written and organized. I would suggest a minor point to further improve the manuscript. As ICIs are currently approved for the first-line systemic treatment of hepatocellular carcinoma (HCC) treatment and most HCC patients have an underlying cirrhosis, the development of hepatoxicity is of paramount clinical relevance. Recent studies have reported a crucial role of CD4+CD25+Foxp3+ regulatory T cells (Tregs) in the pathogenesis of ICI-related hepatotoxity as this immune cell population largely expresses molecular targets of ICIs, as previously described (CTLA-4 control over Foxp3+ regulatory T cell function. Science. 2008;322:271–275; doi: 10.3748/wjg.v27.i22.2994; doi: 10.1126/science.1160062; doi: 10.1007/s00262-019-02427-4; doi: 10.1016/j.semcancer.2019.01.006.). Of relevance, Treg frequency and function is a determinant factor affecting the disease-progression of the underlying chronic liver disease as well as the risk of HCC development/spread and ICI-related hepatotoxicity (doi: 10.3748/wjg.v27.i22.2994) and future clinical studies should explore the pattern of regulatory T lymphocytes during treatment to correlate the development of adverse events with Treg temporal trend.
Thank you for your comment. We implemented this point in the last paragraph (future direction) to
point out its relevance.

Reviewer 2 Report
Comments and Suggestions for Authors
The authors have produced a comprehensive review on immune-mediated hepatotoxicity from checkpoint inhibitors. It is generally well-written and authoritative, but several comments are appropriate:
1. I was not able to view Figure 1 that mentions current guidelines - if this is not a table that provides the most up to date guidance from FDA, EMA, and the various oncology societies - I suggest that they all be included
2. The authors need to provide more detail on delayed hepatotoxicity since PD-1 agents can have effects lasting out to 60-90 days after the last dose - which may confound the diagnosis of irH when other agents are added.
3. The authors should consider adding a section that deals with dealing with hepatotoxicity from new ICIs under development - an important article in this regard is by Regev et al (PMID 32768244) that also deals with rechallenging patients after irH is treated.
4. the introduction mentions a global study that reports a mortality rate of 6% from irH. This seems rather high given that acute liver failure is rare. The authors need to review the methodology of this registry and comment on the causality of the deaths - were they actually due to irH that went on to acute liver failure (due to suboptimal management with immunosuppression or other reasons) or were the deaths due to the underlying malignancy?
5. In my experience, oncologists seem to be quite capable of recognizing irH (and other immune-mediated toxicities) without the need to resort to as complete a workup as the authors recommend. Often, steroids are simply started empirically for typical cases and the diagnosis is confirmed if the patient improves as expected. Can the authors comment on how often oncologists are following the guidance to the letter?
6. I noticed several sentences where the English usage can be improved:
line 463 - Herein (rather than hereby)
line 545 : change to: The use of ICIs is rapidly spreading as they represent the standard of care"
line 577: current role (instead of currently)
line 629: selective cases (instead of selected)
overall the English was very readable.
Comments on the Quality of English Language
see above #6
Author Response
The authors have produced a comprehensive review on immune-mediated hepatotoxicity from checkpoint inhibitors. It is generally well-written and authoritative, but several comments are appropriate:
- I was not able to view Figure 1 that mentions current guidelines - if this is not a table that provides the most up to date guidance from FDA, EMA, and the various oncology societies - I suggest that they all be included
We apologise for this problem. The figure now represents a comprehensive summary of the current guidelines in a readable format.
- The authors need to provide more detail on delayed hepatotoxicity since PD-1 agents can have effects lasting out to 60-90 days after the last dose - which may confound the diagnosis of irH when other agents are added.
Thank you for your comment. More details on delayed toxicity have been included as requested.
- The authors should consider adding a section that deals with dealing with hepatotoxicity from new ICIs under development - an important article in this regard is by Regev et al (PMID 32768244) that also deals with rechallenging patients after irH is treated.
Thank you for your advice. an additional section on hepatotoxicity of ICIs under development has now been included with a specific reference to the Regev et al study.
- the introduction mentions a global study that reports a mortality rate of 6% from irH. This seems rather high given that acute liver failure is rare. The authors need to review the methodology of this registry and comment on the causality of the deaths - were they actually due to irH that went on to acute liver failure (due to suboptimal management with immunosuppression or other reasons) or were the deaths due to the underlying malignancy?
The study mentioned in the introduction refers to a comprehensive worldwide pharmacovigilance study based on data extracted from VigiBase, the international pharmacovigilance database, from 2008 until 01/2023. The reported death rate data are correct, however it must be pointed out the existence of a report specifying that the causes of registered death might not all be related to irH (supplementary figure 6). We would like to thank you for this report and we have now provided a specific comment t on the issue.
- In my experience, oncologists seem to be quite capable of recognizing irH (and other immune-mediated toxicities) without the need to resort to as complete a workup as the authors recommend. Often, steroids are simply started empirically for typical cases and the diagnosis is confirmed if the patient improves as expected. Can the authors comment on how often oncologists are following the guidance to the letter?
Thank you for the comment. The diagnostic work-up reported in our manuscript, reflects what suggested by the most recent guidelines. To the best of our knowledge, there are no such data describing the rate of physician’s (oncologist) adherence to guidelines. Indeed, this should be a relevant point for further researches
- I noticed several sentences where the English usage can be improved:
line 463: Herein (rather than hereby)
line 545: change to: “The use of ICIs is rapidly spreading as they represent the standard of care"
line 577: current role (instead of currently)
line 629: selective cases (instead of selected)
overall the English was very readable.
Thank you, all the requested corrections have been made, together with a more detailed English proofreading.
